

# A normalized autoencoder for LHC triggers

**Barry M. Dillon[1], Luigi Favaro[1], Tilman Plehn[1], Peter Sorrenson[2] and Michael Krämer[3]**

**1** Institut für Theoretische Physik, Universität Heidelberg, Germany
**2** Heidelberg Collaboratory for Image Processing, Universität Heidelberg, Germany
**3** Institute for Theoretical Particle Physics and Cosmology (TTK),
RWTH Aachen University, Germany

## Abstract

Autoencoders are an effective analysis tool for the LHC, as they represent one of its main goal of finding physics beyond the Standard Model. The key challenge is that out-of-distribution anomaly searches based on the compressibility of features do not apply to the LHC, while existing density-based searches lack performance. We present the first autoencoder which identifies anomalous jets symmetrically in the directions of higher and lower complexity. The normalized autoencoder combines a standard bottleneck architecture with a well-defined probabilistic description. It works better than all available autoencoders for top vs QCD jets and reliably identifies different dark-jet signals.

## Contents

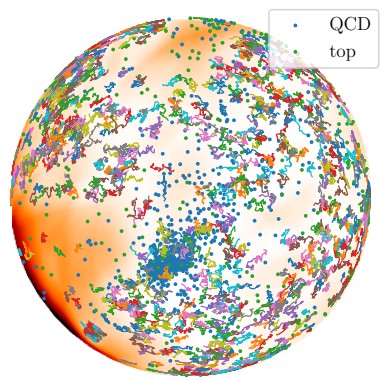

# 1 Introduction

The big goal of the LHC is to discover physics beyond the Standard Model (BSM) and to identify new properties of the fundamental constituents of matter. Until now, we pursue BSM searches based on pre-defined theory hypotheses. The upcoming LHC runs need to supplement targeted searches with (i) analyses of phase space regions linked to an effective theory extension of the Standard Model and (ii) searches for anomalous effects defined as not explained by the Standard Model. Both of these strategies share the ambitious goal of understanding all aspects of LHC data using fundamental physics — with the secret goal of bringing down the Standard Model.

A key concept in the search for anomalies in LHC data is that as few as possible assumptions should be made about the potential signals in the data. The techniques used for this are strongly influenced by developments in modern machine learning (ML). Autoencoders (AEs) are simple tools for anomaly searches, based on a bottleneck in the mapping of a data representation onto itself. They can, for instance, identify anomalous jets in a QCD jet sample [1,2]. Because the bottleneck as the latent space does not have a well-defined structure, the anomaly score has to be related to the quality of the reconstruction.

Adding a latent space structure leads us to more complex tools, for example variational autoencoders (VAEs) [3]. In the encoding step for a VAE we map a high-dimensional data representation to a low-dimensional latent distribution; the decoding step then generates new high-dimensional objects. The latent space will encode structures which might not be apparent in the high-dimensional input data. Such VAEs work for anomaly searches with LHC jets [4,5], and we can trade the reconstruction loss for an anomaly score defined in latent space. In this case, we benefit from a network architecture which constructs an optimized latent space, for instance the Dirichlet VAE [6], leading to a mode separation between background and signal.

Motivated by their initial success, ML-methods for anomaly detection at the LHC were developed for anomalous jets [7–16], anomalous events [17–35], or to enhance search strategies [36–44]. They include a first ATLAS analysis [45], experimental validation [46, 47], quantum machine learning [48], self-supervised learning [49, 50], applications to heavy-ion collisions [51], the DarkMachines community challenge [52], and the LHC Olympics 2020 community challenge [53, 54].

The problem with these studies is that it is not clear what the anomalous property of jets or events actually means. The step from AEs to VAEs implies a change in the way we define anomalous jets. For an unstructured AE we search for out-of-distribution jets based

on the compressibility of their features. The lack of symmetry in tagging anomalous top jets vs anomalous QCD jets in the respective other sample casts doubts on this definition for the LHC. While massive top jets stick out among low-mass QCD jets, we cannot expect a QCD jet to be special in a sample of top jets [55] since its simpler features will be interpolated by the neural network. A better-suited definition is based on low-probability regions in the background phase space distributions [56–60]. Here we can encode the phase space density, for instance, using cluster algorithms, VAEs, or a normalizing flow [15], but none of these methods are especially successful at identifying anomalous jets once the signal becomes more challenging than top jets.

In this paper we present the normalised autoencoder (NAE) [61] as a remedy for these issues. It relies on detecting outliers through the reconstruction loss, but via an energy-based model [62, 63]. One of the goals of this work is to develop an autoencoder which is a robust anomalous jets tagger. We explore the concept of using autoencoders as of triggers, i.e. tools that can extract interesting events from a given background with as little bias as possible. Although the jets we study in the paper would pass trigger selection cuts already, the results still demonstrate how our approach limits the assumptions made on BSM signals to data preprocessing rather than latent space structure, in favor of a more model-agnostic network architecture. The NAE achieves reliable anomaly detection results without increasing the size of the network, with the additional components affecting only how the model is trained. The training focuses on minimizing the negative log-likelihood of the data given the network parameters, but evaluating a probability distribution. This means that NAE is a probabilistic model which samples from the model distribution and penalizes modes absent from the training data. For phase space regions with such modes the NAE training adjusts the energy as the underlying structure of the latent space, such that the autoencoder gets a robust OOD detector. Two Langevin Markov Chains in the latent and phase spaces probe the poor-reconstruction regions and penalize them. Combining this with the minimization of the training reconstruction error, we define a minmax loss function that converges when the training and model distributions match each other.

In Sec. 2 we first introduce energy-based models as an alternative to reconstruction losses like an MSE or a likelihood ratio. We then describe the NAE setup,[1] with its efficient way of sampling the background data manifold in phase space and latent space. In Sec. 3 we apply the NAE to the top tagging dataset [64–66] and show that, for the first time, the NAE identifies anomalous top jets and anomalous QCD jets symmetrically and with high efficiency. Next, we target two challenging dark jet signals [15] and confirm the excellent performance of the NAE and its relative independence of the jet image preprocessing in Sec. 4. In the Appendix we provide additional details about the NAE and our implementation.

# 2 Network and dataset

The normalized autoencoder [61] we will use for this study is an energy-based modification of a standard AE, as applied in Ref. [1]. We will first introduce energy-based models, mention their challenges, and then describe the way the NAE modifies the AE training. As input data we use jet images with standard preprocessing.

## 2.1 Energy-based networks

Energy-Based Models (EBMs) are a class of probability density estimation models appealing for their flexibility. They are defined through a normalizable energy function, which is minimized

---

[1]The code is available at https://github.com/heidelberg-hepml/normalized-autoencoders

during training. This energy function can be chosen as any non-linear function mapping a point to a scalar value [67],

$$E_\theta(x) \colon \mathbb{R}^D \to \mathbb{R}, \tag{1}$$

where $D$ is the dimensionality of the phase space. The EBM uses this energy function to define a probabilistic loss, assuming a Boltzmann or Gibbs distribution as its probability density over phase space,

$$p_\theta(x) = \frac{e^{-E_\theta(x)}}{Z_\theta}, \qquad \text{with} \qquad Z_\theta = \int_x dx\, e^{-E_\theta(x)}, \tag{2}$$

with the partition function $Z_\theta$. We omit an explicit normalization of the energy by a temperature or some other constant in this formula. The main feature of a Boltzmann distribution is that low-energy states have the highest probability. The EBM loss is the negative logarithmic probability evaluated as a likelihood over the model parameters,

$$\mathcal{L}(x) = -\log p_\theta(x) = E_\theta(x) + \log Z_\theta \qquad \Rightarrow \qquad \mathcal{L} = \left\langle E_\theta(x) + \log Z_\theta \right\rangle_{x \sim p_{\text{data}}}, \tag{3}$$

where we define the total loss as the expectation over the per-sample loss. The difference to typical likelihood losses is that the second, normalization term is unknown.

To train the network we want to minimize the loss in Eq.(3), so we have to compute its gradient,

$$
\begin{aligned}
\nabla_\theta \mathcal{L}(x) = -\nabla_\theta \log p_\theta(x) &= \nabla_\theta E_\theta(x) + \nabla_\theta \log Z_\theta \\
&= \nabla_\theta E_\theta(x) + \frac{1}{Z_\theta} \nabla_\theta \int_x dx\, e^{-E_\theta(x)} \\
&= \nabla_\theta E_\theta(x) - \int_x dx\, \frac{e^{-E_\theta(x)}}{Z_\theta} \nabla_\theta E_\theta(x) \\
&= \nabla_\theta E_\theta(x) - \left\langle \nabla_\theta E_\theta(x) \right\rangle_{x \sim p_\theta}.
\end{aligned} \tag{4}
$$

The first term in this expression can be obtained using automatic differentiation from the training sample, while the second term is intractable and must be approximated. Computing the expectation value over $p_{\text{data}}(x)$ allows us to rewrite the gradient of the loss as the difference of two energy gradients

$$\left\langle \nabla_\theta \mathcal{L}(x) \right\rangle_{x \sim p_{\text{data}}} = \left\langle -\nabla_\theta \log p_\theta(x) \right\rangle_{x \sim p_{\text{data}}} = \left\langle \nabla_\theta E_\theta(x) \right\rangle_{x \sim p_{\text{data}}} - \left\langle \nabla_\theta E_\theta(x) \right\rangle_{x \sim p_\theta}. \tag{5}$$

The first term samples from the training data, the second from the model. According to the sign of the energy in the loss function, the contribution from the training dataset is referred to as positive energy and the contribution from the model as negative energy. One way to look at the second term is as a normalization which ensures that $\mathcal{L} = 0$ for $p_\theta(x) = p_{\text{data}}(x)$. Another way is to view it as inducing a structure for the minimization of the likelihood.

One practical way of sampling from $p_\theta(x)$ is to use Markov-Chain Monte Carlo (MCMC). We use Langevin Markov Chains, where the steps are defined by drifting a random walk towards high probability points according to

$$x_{t+1} = x_t + \lambda_x \nabla_x \log p_\theta(x) + \sigma_x \epsilon_t, \qquad \text{with} \qquad \epsilon_t \sim \mathcal{N}_{0,1}. \tag{6}$$

Here, $\lambda$ is the step size and $\sigma$ the noise standard deviation. When $2\lambda = \sigma^2$ the equation resembles Brownian motion and gives exact samples from $p_\theta(x)$ in the limit of $t \to +\infty$ and $\sigma \to 0$.

For ML applications working on images, the high dimensionality of the data makes it difficult to cover the entire physics space $x$ with Markov chains of reasonable length. For this reason, it is common to use shorter chains and to choose $\lambda$ and $\sigma$ to place more weight on the gradient term than on the noise term. If $2\lambda \neq \sigma^2$, this is equivalent to sampling from the distribution at a different temperature defined as

$$T = \frac{\sigma^2}{2\lambda}. \tag{7}$$

There the two parameters are defined in Eq.(6). By upweighting $\lambda$ or downweighting $\sigma$ we are effectively sampling from the distribution at a low temperature, thereby converging more quickly to the modes of the distribution.

By inspecting the expectation value of the loss in the form of Eq.(5), we can identify the training as a minmax problem, where we minimize the energy of the training samples and maximize the energy of the MCMC samples. This means that the energy of training data points is pushed downwards. At the same time the energy of Markov chains sampled from the energy model distribution will be pushed upwards. For instance, if $p_\theta(x)$ reproduces $p_{\text{data}}(x)$ over most of the phase space $x$, but $p_\theta(x)$ includes an additional mode, its phase space region will be assigned large values of $E_\theta(x)$ through the minimization of the loss. This way, all modes present in the energy model distribution but missing from the training distribution $p_{\text{data}}$ will be suppressed. This process of adjusting the energy continues until the model reaches the equilibrium in which the model distribution is identical to the training data distribution.

Despite the well-defined algorithm, training EBMs is difficult due to instabilities arising from (i) the minmax optimization, with similar dynamics to balancing a generator and discriminator in a GAN; (ii) potentially biased sampling from the MCMC due to a low effective temperature; and (iii) instabilities in the LMC chains. Altogether, stabilizing the training during its different phases requires serious effort.

## 2.2 Normalized autoencoder

An AE is a two-module function that maps an input to its reconstruction using an encoder-decoder structure,

$$f_\theta(x): \quad \mathbb{R}^D \longrightarrow \mathbb{R}^{D_z} \xrightarrow{f_D} \mathbb{R}^D. \tag{8}$$

The training minimizes the per-pixel difference between the original input and its mapping. A typical choice for the loss function is the Mean Squared Error (MSE) of this reconstruction. With this definition a plain AE is not a probabilistic model, since a small reconstruction error does not correspond to a large likelihood. However, we can upgrade the AE to a probabilistic NAE by using the MSE as the energy function in Eq.(1)

$$E_\theta(x) = \text{MSE} \equiv \frac{1}{N} \sum_{\text{pixels}} |x - f_\theta(x)|^2. \tag{9}$$

This way we can train a probabilistic AE using Eqs.(5) and (6).

The NAE training includes two steps. First, we pre-train the baseline AE with the standard MSE loss, similar to Ref. [1]. After the AE pre-training we switch to the NAE loss given in Eq.(3). All NAE parameters are given in Tab. 1. In the spirit of a proper anomaly search we use the same network and hyper-parameters throughout this paper. They reflect a trade-off between sampling quality, training stability, and tagging performance.

The pre-training phase builds an approximate density estimator exclusively based on the training data by minimizing the reconstruction error. Then, the NAE loss explores the regions with low energy and guarantees the behavior of the model especially in the region close to

but not in the training data distribution. Here, the mismatch between the data and the model distribution is corrected by the inter-play between the two components of the loss function. We cannot give such a guarantee for a standard autoencoder, which only sees the training distribution and could assign arbitrary reconstruction scores to data outside this distribution. In fact this is the source of the problems with standard autoencoders outlined in the introduction, which are solved using the NAE.

As mentioned above, training EBMs is a practical challenge. Several algorithms have been proposed to train these networks. Two well-known methods based on MCMC samples are Contrastive Divergence (CD) and Persistent CD. CD and PCD differ in how they define the initial sample. CD uses a sample taken from the data distribution $p_{\text{data}}(x)$ while PCD samples from a replay buffer made up of the final state of Markov chains from previous steps of the optimization. However, these methods are susceptible to creating spurious high density modes and struggle with full space coverage [61].

We follow a different approach, using the fact that we can train a regular autoencoder before starting the NAE training. If we accept that different initializations of the MCMC defined in Eq.(6) lead to different results, we can tune $\lambda_x$ and $\sigma_x$ in such a way that we can use a sizeable number of short, non-overlapping Markov chains [63, 68, 69]. Specifically, the proposed algorithm for an efficient training of NAEs is On-Manifold Initialization (OMI) [61]. This approach is motivated by the observation that sampling the full data space is inefficient due to its high dimensionality, but the training data lies close to a low-dimensional manifold embedded in the data space $x$. All we need to do is to sample close to this manifold. Since we are using an autoencoder this manifold is defined implicitly as the image of the decoder network, meaning that any point in the latent space $z$ passed through the decoder will lie on the manifold. This means we can first focus on the manifold by taking samples from a suitably defined distribution in the low-dimensional latent space, and then map these samples into data space via the decoder. After that, we perform a series of MCMC steps in the full ambient data space to allow the Markov chains to minimize the loss around the manifold.

To sample from the model we first need to define a suitable latent probability density, which we do as

$$q_\theta(z) = \frac{e^{-H_\theta(z)}}{\Psi_\theta}, \qquad \text{with} \qquad H_\theta(z) = E_\theta(f_D(z)), \tag{10}$$

where $H_\theta(z)$ is the latent energy, and $f_D$ is the decoder network, all in complete analogy to Eq.(2). Having defined these quantities, the latent space chain is run as

$$z_{t+1} = z_t + \lambda_z \nabla_z \log q_\theta(z) + \sigma_z \epsilon_t, \qquad \text{with} \qquad \epsilon_t \sim \mathcal{N}_{0,1}. \tag{11}$$

Once we reach a high-density point on the decoder manifold, the final sample is obtained by running a second input chain according to Eq.(6).

During the OMI it is crucial that we cover the entire latent space, thus a compact structure is preferable. To achieve that, we normalize the latent vectors so that they lie on the surface of a hypersphere $\mathbb{S}^{D_z-1}$, allowing for a uniform sampling of the initial batch in the latent space. The step size and the noise of both chains are tuned to give $T < 1$. Even if a lower temperature introduces a bias towards modes with lower reconstruction error, this helps stabilize the training and obtain finer samples from the MCMC. Long LMC chains are affected by instability by two reasons: sudden changes in the gradients between steps, and diverging energy for both positive and negative samples due to the loss function being independent of constant shifts. Even if these issues are still not well understood in the ML community, different possible solutions have been proposed [63] and applied in this work: (i) clipping gradients in each step; (ii) spectral normalization; (iii) L2 weight normalization; and (iv) L2 normalization on positive and negative samples. To decrease the size of both chains, a replay buffer has been utilized which saves the final points of each latent chain. In the next iteration the initial sample is

either drawn from the buffer or drawn uniformly from the hypersphere, with the probability of being drawn from the buffer being 0.95. Finally, an acceptance Metropolis step and a noise annealing step can be applied.

The encoder has 5 convolutional layers. Each layer has 8 filters, except for the last layer with one filter. The output is then flattened, and two dense layers downsize the network to the latent space size. The decoder mimics the encoder with 2 dense layers followed by 4 convolutional layers. All intermediate activation functions are PReLU. The output activation for the encoder and the decoder are linear and sigmoid, respectively. For the latent space dimension we use $D_z = 3$, which is not optimized for performance, but allows us to visualize the latent space easily. We run the pre-training for 300 epochs, using Adam [70] with default parameters. Additional information on the network architecture can be found in A

## 2.3 Jet Images

As our application we choose jet images. The first test will be top vs QCD, discussed in Sec. 3, where we use the top-tagging dataset [64–66], also used for the AE in Ref. [1] and the Dirichlet VAE in Ref. [6]. We start with anti-$k_T$ jets [71] with $R = 0.8$, defined by FASTJET3.1.3 [72] as substructure containers. The top and QCD jets are are required to have

$$p_T = 550 \ldots 650 \text{ GeV}, \qquad \text{and} \qquad |\eta| < 2. \tag{12}$$

Before defining the jet image, the jets are pre-processed by centering each jet around the $k_T$-weighted centroid of all constituents. Then, the jets are rotated such that their principal axis points to 12 o'clock, and flipped so that the highest $p_T$ region is in the lower-left quadrant. Then, the constituents are pixelized in $40 \times 40$ images with pixel size $[\Delta\eta, \Delta\phi] = [0.029, 0.035]$. The intensity of the pixels is defined by the sum of $p_T$s within that cell, and finally, the whole image is rescaled by the total $p_T$ of the event. To reduce the sparsity we apply a Gaussian filter to each image. The effect of two Gaussian filters is illustrated in Fig. 1. Our top tagging dataset consists of 140k jets for each class, of which we use 100k jets for training, and the remaining 40k for testing. The jet images are pre-processed with a Gaussian filter with $\sigma_G = 1$.

Our second reference dataset, presented in Sec. 4 are two dark-matter-inspired signal samples [15]. The underlying model is hidden valleys, with a light and strongly interacting dark sector [73–75]. Particles produced in this dark sector can decay within that sector and form a dark shower. Such a dark shower will eventually switch to SM-fragmentation and form either a semi-visible jet [76–81] or a pure, modified QCD jet [1]. We will refer to the semi-visible jets

Table 1: LMC and training parameters. The temperature is implicitly fixed by the noise and the step size as $T_x = 10^{-7}$ and $T_z \approx 10^{-6}$.

| LMC parameters | latent | input |
|---|---|---|
| $\lambda$ | 100 | 50 |
| $\sigma$ | $10^{-2}$ | $10^{-4}$ |
| # of steps | 30 | 30 |
| metropolis | ✓ | ✓ |
| annealing | – | ✓ |
| training parameters | pre-AE | NAE |
| learning rate | $10^{-3}$ | $10^{-5}$ |
| iterations | 15k | 40k |
| batch size | 2048 | 128 |

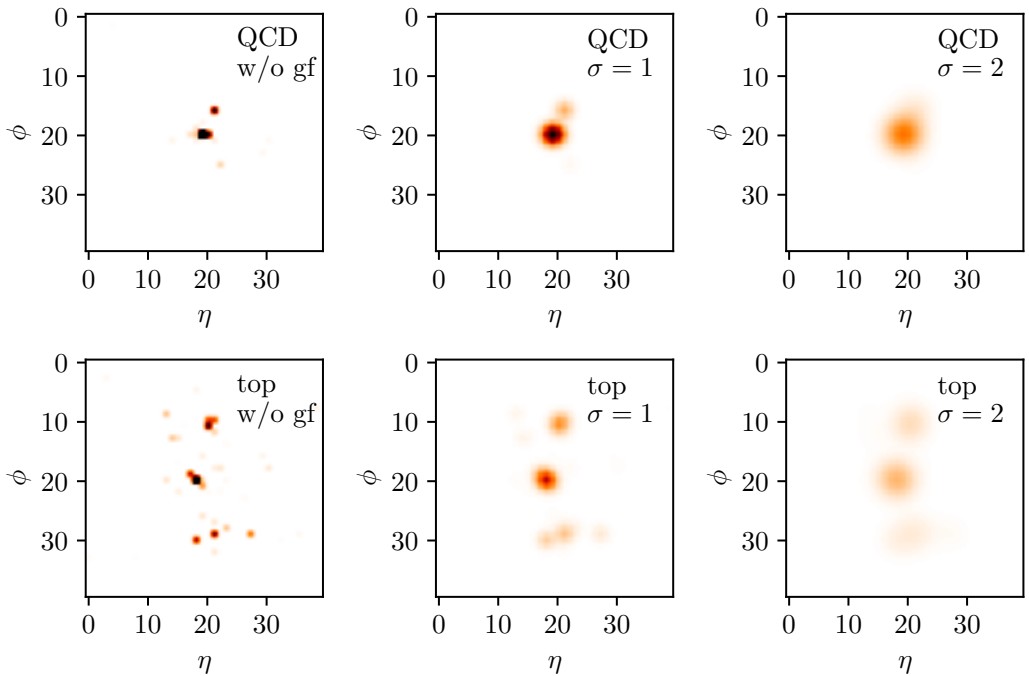

Figure 1: Example QCD and top images without Gaussian filter, $\sigma_G = 1$, and $\sigma_G = 2$.

as the Aachen dataset and the modified QCD jets as the Heidelberg dataset. Compared to the QCD background, the Aachen dataset is mostly more sparse, whereas the Heidelberg dataset includes an additional decay structure. The two signal samples and the QCD background sample are generated just like to the top jet sample, but with

$$p_{T,j} = 150 \ldots 300 \text{ GeV}, \qquad \text{and} \qquad |\eta_j| < 2. \tag{13}$$

As for the top jets, a Gaussian filter improves the network training and we use a filter with a $\sigma_G = 1$, but some additional precautions are needed for dark jets. We know that for an efficient identification of both of the dark jets we need to reweight the jet images. Unlike in Ref. [15] we now apply the same pixel-wise remapping for both dark jet signals, namely

$$p_T \quad \rightarrow p_T^n, \qquad \text{with} \qquad n = 0.01, 0.1, 0.2, 0.3, 0.5. \tag{14}$$

The goal is to reduce the dependence of the autoencoder performance on this remapping for different signals.

# 3 QCD vs top jets

A simple, established anomaly detection task based on jet images is to extract top jets out of a QCD jet sample, with network training on background only [1]. We summarize the results with a focus on the performance for symmetric tagging of QCD vs top images. In standard autoencoder models a known problem is that they tend to assign larger reconstruction losses to samples with higher complexity rather than those which are not well-represented in the training data. This is exemplified when training an autoencoder on QCD jets to identify anomalous top jets, versus training the autoencoder on top jets to identify anomalous QCD jets. The underlying physics suggests that it should be easy to find large regions of phase space exclusively

populated by QCD or top jets, therefore allowing for out-of-distribution detection in both directions. However, the autoencoder works well in the direction of anomalous top jets, while it does not work well in the direction of anomalous QCD jets. We refer to any anomaly detection technique that tags in both directions of higher and lower jet complexity, without modifications in the architecture and training, as symmetric.

A known issue in training EBMs is a potential collapse of the sampler, detected by a diverging negative energy and a collapse of the sampled images [63,82]. To find a sweet spot between mode coverage and stability requires careful tuning of the LMC parameters, in addition to a regularization. We only encounter this failure when training on top jets, because the latent space undergoes drastic changes. To detect a collapse, we use several diagnostic tools. A proper training shows stable positive and negative energies, a fluctuating loss function close to zero, and smooth variations of the weights. In addition, we can directly look at the sampled images saving batches after a fixed number of iterations. The NAE training is carried over for 50 epochs or until a collapse of the sampler happens. Then, the best model is chosen by taking the iteration with the loss function closest to zero and with stable positive and negative energies.

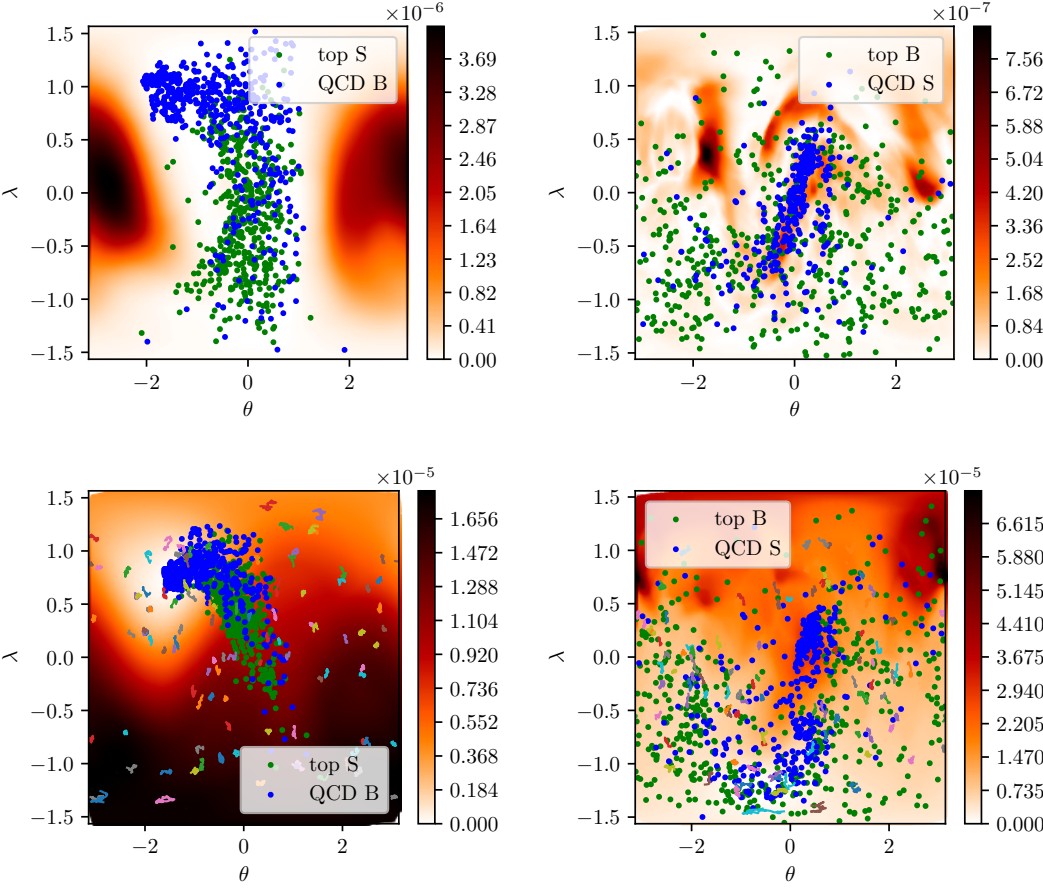

Figure 2: Equirectangular projection of the latent space after pre-training (upper) and after NAE training (lower). The $x$- and $y$-axis are the longitude and latitude on the sphere. We train on QCD jets (left) and on top jets (right). The background color indicates the energy over the latent space, the lines represent the path of the LMCs in the current iteration, and the points show the positions of jets from both samples.

We choose a three-dimensional latent space for our model, making it a sphere in three dimensions. We exploit this low dimensionality to visualize the development of the latent energy landscape. We employ an equirectangular projection as shown in Fig. 2. The $x$-axis and the $y$-axis give the longitude and the latitude on the sphere. To reduce the distortion around jets, the poles are chosen such that the center $(0, 0)$ is given by the region with most jets. By sampling points on the sphere and calculating the energy of the decoded jets we build the latent landscapes. In this landscape we show the path of latent LMCs and the position of encoded jets from both distribution.

In the upper panels of Fig. 2 we show a projection of the latent space after minimization of the MSE, like in the usual AE, but using a compact, spherical latent space. In the left panels we train on the simpler QCD background, which means that the latent space has a simple structure. The QCD jets are distributed widely over the low-energy region, while the anomalous top jets cluster slightly away from the QCD jets. The situation changes when we train on the more complex top jets, as shown in the right panels. The latent MSE or energy-landscape reflects this complex structure with many minima, and top jets spread over most of the sphere. After the NAE training, only the regions populated by training data have a low energy. The sampling procedure has shaped the decoder manifold to correctly reconstruct only training jet images. For both training directions, the Markov chains move from a uniform distribution to mostly cover the region with low energy, leading to an improved separation of the respective backgrounds and signals.

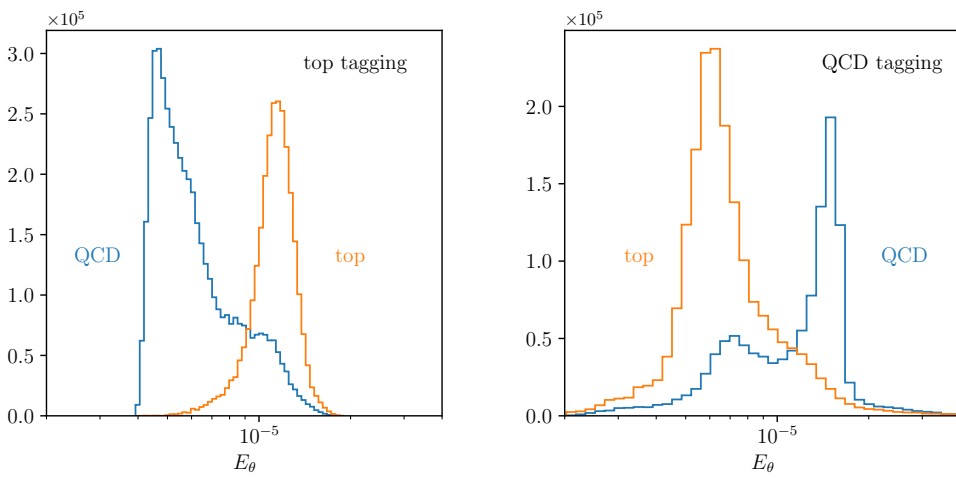

Figure 3: Distribution of the energy or MSE after training on QCD jets (left) and on top jets (right). We show the energy for QCD jets (blue) and top jets (orange) in both cases.

To see the difference in the two-directional training we can also look at the respective energy distributions. In the left panel of Fig. 3 we first see the result after training the NAE on QCD jets. The energy values for the background are peaked strongly, cut off below $4 \times 10^{-5}$ and with a smooth tail towards larger energy values. The energy distribution for top jets is peaked at larger values, and again with an unstructured tail into the QCD region. We can then evaluate the performance of anomalous top tagging in terms of the ROC curve, the AUC score, and the inverse mistag at low efficiency ($\epsilon_s = 0.2$) in Fig. 4. This choice of working point is motivated by possible applications of autoencoders requiring significant background rejection. The orange ROC curves show how the performance increases after the additional AE training to the NAE training in the self-constructed latent-space geometry. The AUC value of 0.91 quoted in the corresponding table is above the AE setup and our earlier studies.

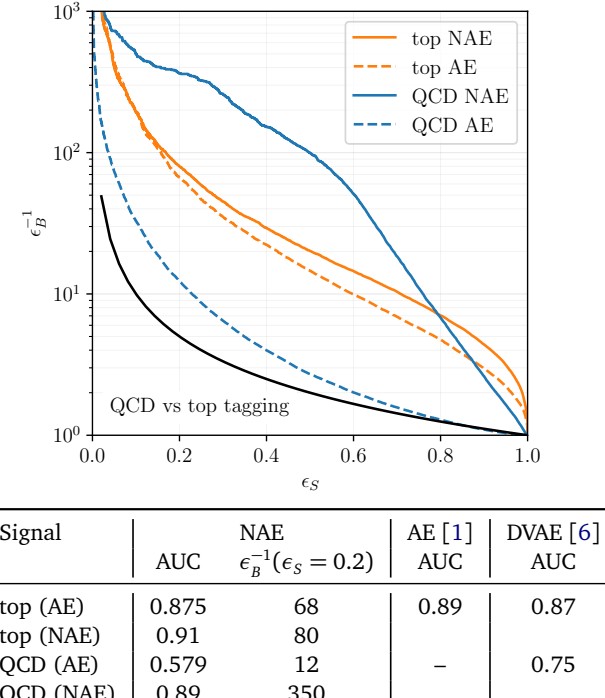

| Signal | | NAE | AE [1] | DVAE [6] |
| | AUC | $\epsilon_B^{-1}(\epsilon_S = 0.2)$ | AUC | AUC |
| --- | --- | --- | --- | --- |
| top (AE) | 0.875 | 68 | 0.89 | 0.87 |
| top (NAE) | 0.91 | 80 | | |
| QCD (AE) | 0.579 | 12 | – | 0.75 |
| QCD (NAE) | 0.89 | 350 | | |

Figure 4: ROC curve for top (orange) and QCD (blue) tagging after AE pre-training (dashed), and after NAE training (solid). A random classifier corresponds to the solid black line. In the table we compare the performance of the NAE, and the pre-trained AE used here, to two studies in the literature.

Next, we can see what happens when we train on top jets and search for the simpler QCD jets as an anomaly. In the right panel of Fig. 3 the background energy is much broader, with a significant tail also towards small energy values. The QCD distribution develops two distinct peaks, an expected peak in the tail of the top distribution and an additional peak under the top peak. The fact that the NAE manages to push the QCD jets towards larger energy values indicates that the NAE works beyond the compressibility ordering of the simple AE. However, the second peak shows that a fraction of QCD jets look just like top jets to the NAE. The ROC curves in Fig. 4 first confirm that training a regular AE to search for QCD jets in a top sample makes little sense, leading to an AUC value of 0.579. After the additional NAE training step we reach an ROC value of almost 0.9, close to the corresponding value for top tagging. However, the shape of the ROC curve does not exactly follow our expectations. We can start with large $\epsilon_S \to 1$ in the right panel of Fig. 3. Here the working point is in the small-energy tails of the signal and background distributions, and because of the tails in the top jet distribution the performance of the classification network starts poorly. Moving towards smaller $\epsilon_S$ the network performance drastically improves, until we pass the background peak, corresponding to $\epsilon_S \sim 0.6$. Below this value, the QCD tagging improves, again, but more slowly than the corresponding top tagging.

Altogether, we see in the right panels of Fig. 4 that the NAE combines competitive performance with symmetric tagging top and QCD tagging. In the easier direction of top tagging it beats the AE and DVAE benchmarks in spite of the non-optimized setup, and in the reverse direction of QCD tagging it provides competitive results for the first time.

# 4 QCD vs dark jets

After testing NAE on this benchmark process, we can move to a more difficult task, namely tagging two distinct kinds of dark jets with the same network. The signal datasets are the same as in Ref. [15].

To first illustrate the $p_T$-reweighting we select the most poorly reconstructed 1000 QCD images, according to their MSE or energy. In Fig. 5 we show the average of these images to the left, the average reconstruction in the second column, and the pixel-wise energy between the two in the third row. Reducing the remapping defined in Eq.(14) from $n = 0.5$ to $n = 0.01$ washes out the $p_T$-structures, so the input and especially the reconstructed images change from more structured jets to a simple, single-prong structure. For our two signal hypotheses this means that for large $n$ the poorly reconstructed QCD images resemble the Heidelberg signal, leading to a more efficient signal extraction, while for small $n$ the poorly reconstructed jet images resemble the Aachen dataset.

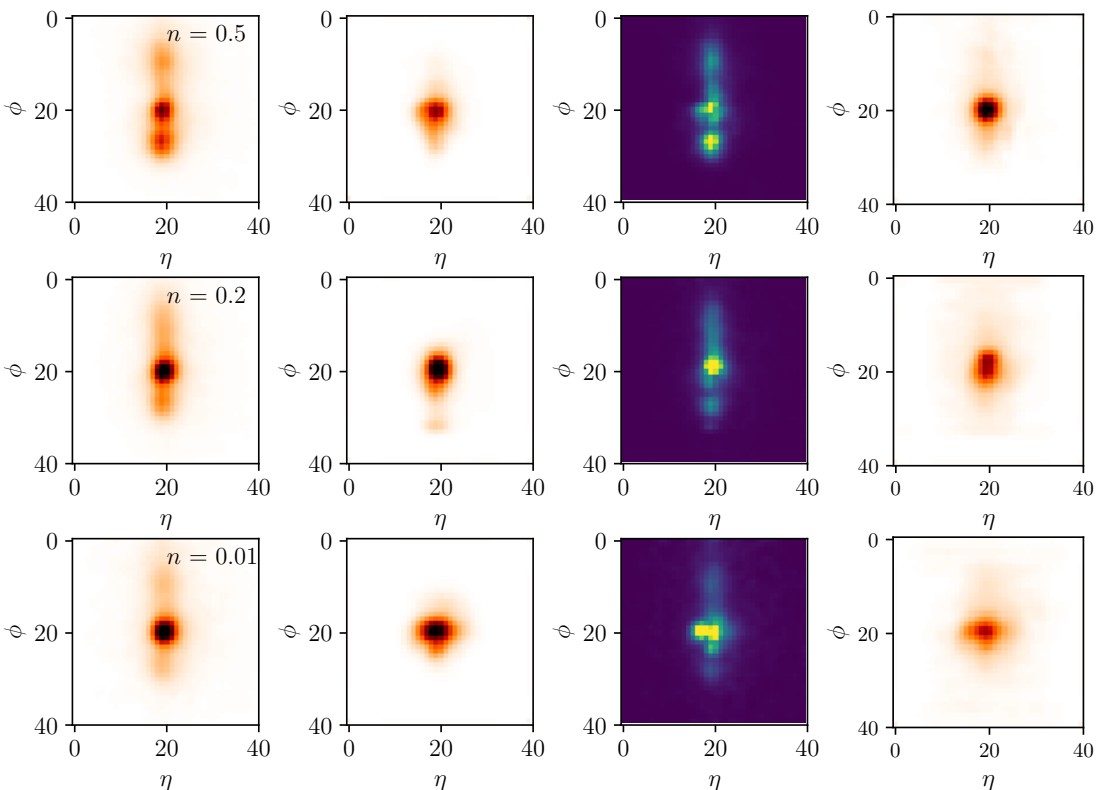

Figure 5: Average QCD jet images for the 1k most poorly reconstructed jets, from left to right: average input, average reconstruction, pixel-wise energy between the two, and average output of the negative energy sample used during training in the last iteration. The rows correspond to reweighting factors $n = 0.5, 0.2, 0.001$.

This difference in the jet reconstruction for different models can be explained by looking at the sampled distributions during training. The NAE-sampled average of the negative-energy jets in the last iteration is shown in the two right column of Fig. 5. At $n = 0.5$ the NAE sampling discards all secondary clusters and focuses on the main feature of the QCD jets, the single prong. During training, the loss function enhances the main feature by increasing the energy of everything around it in the latent and phase spaces. As a consequence, the initial background is lost after some epochs, but to keep the normalization of each jet the central prong is enhanced. As a result, the tagging of two-prongs structure like the Heidelberg

jets is improved. Conversely, at $n = 0.01$ the reweighting enhances the secondary cluster, which cannot be discarded by the training anymore. As a result, in both initial iteration, and at equilibrium there is a residual background away from the central prong. This way, the NAE training increases the energy for Aachen jets, because the normalization forces the main prong to a lower value. These effects are inevitable when training likelihood-based models, a different preprocessing will change the density and, therefore, the anomaly score.

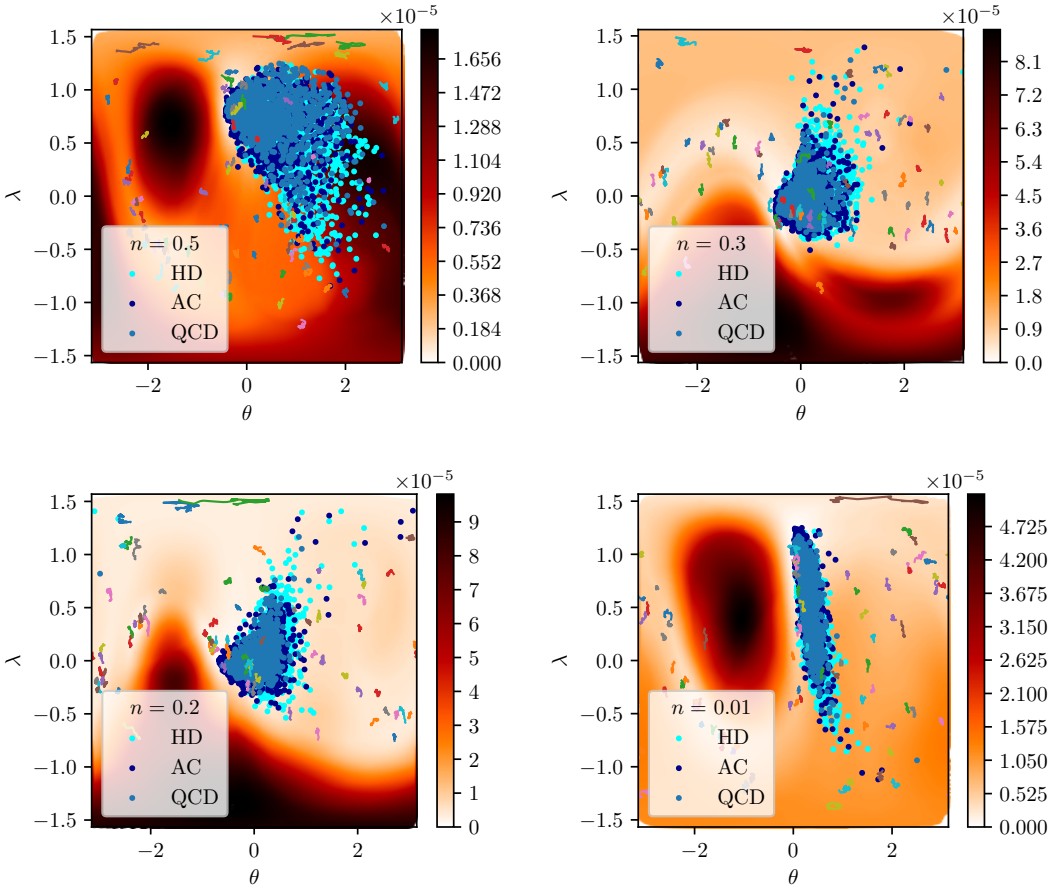

Figure 6: Equirectangular projection of latent spaces after NAE training on QCD jets to identify anomalous dark jets. We show four different $p_T$-reweightings $n$. The blue points represent a sub-sample of QCD training events.

As for the top vs QCD tagging, we then show the latent space landscapes after NAE training in Fig. 6. For all $p_T$-reweightings the network identifies the least populated regions in the decoder manifold and increases the corresponding energy. As discussed above, a large $n = 0.5$ enhances the sensitivity to the more complex Heidelberg dataset, while the sparse Aachen dataset is hardly separated from the QCD jets. For small $n = 0.01$ a distinct region appears at large $\lambda$, where the Aachen signal extends beyond the QCD region. In between the two extremes, the latent landscape changes smoothly with the biggest change happening around $n = 0.2$. Around this point the training can oscillate between focusing on primary prongs or on secondary clusters, causing fluctuations in the performance. This transition in the $p_T$-reweighting is also the only case where the hyperparameters, and especially the temperatures, have a noteworthy effect on the network performance.

Once again focusing on the $p_T$-reweightings we show the energy distributions for the QCD training data and the two signals in Fig. 7. We see that unlike in our earlier study [15] the

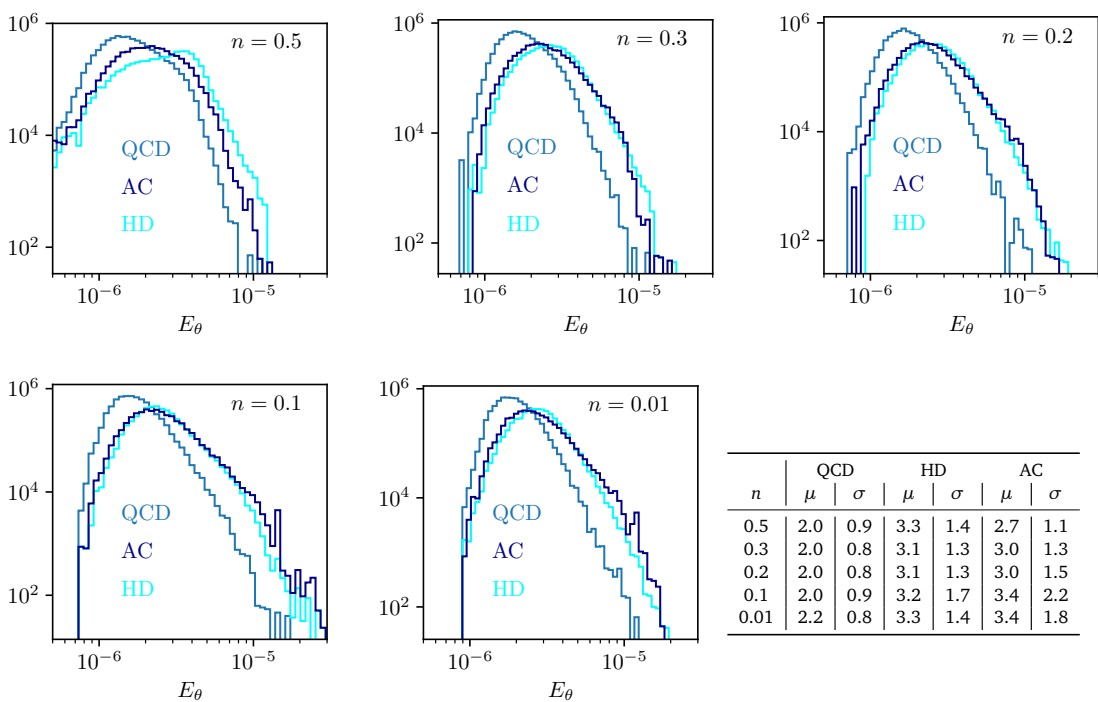

Figure 7: Distribution of the energy for QCD, Aachen, and Heidelberg datasets. Each panel corresponds to a different reweighting of the same datasets. The table shows the mean and the standard deviation for each distribution ($\times 10^{-6}$).

effect of the preprocessing on the whole distribution is limited. A shift in performance at low signal efficiency can be seen by varying $n$ with the ordering between the two dataset being switched around $n = 0.2 \dots 0.3$. The energy distribution of the Heidelberg dataset has a shifted main peak at $n = 0.5$ which is washed out by smaller reweighting factors, while the QCD distribution undergoes a slight shift and develops a longer high-energy tail. For the Aachen dataset, lowering $n$ moves the mean away from the QCD background and at the same time increases the width of the distribution. These patterns will affect the ROC curves at low signal efficiency and large background suppression.

Once we understand how the $p_T$-reweighting changes the input distributions and the energy distributions for the QCD background and the dark jets signals, we can measure the difference between QCD jets and each of the two signals by computing the maximum mean discrepancy (MMD) [83] of sub-samples from these distributions. We show these MMD curves as a function of $n$ in Fig. 8, for the full distributions of 20k QCD and signal jets and only considering the 1k most poorly reconstructed jets in the high-background-rejection target region. For the input distributions we see the same trend in both panels — small $n$ benefits the tagging performance for the Aachen dataset and decreases for the Heidelberg dataset; increasing $n$ improves the tagging performance for the Heidelberg dataset.

The more interesting question is if the input-space pattern can also be observed in the latent-space distributions. The idea behind this test is to construct an autoencoder with a choice of anomaly scores, either reconstruction-based in phase space or in the latent space [6]. Again in Fig. 8 we show the corresponding MMD values as dashed curves. For the complete samples as well as for the most poorly reconstructed jets the latent-space MMD behaves just like the phase-space MMD. This indicates that it should, if necessary, be possible to construct a latent-space anomaly score for the NAE.



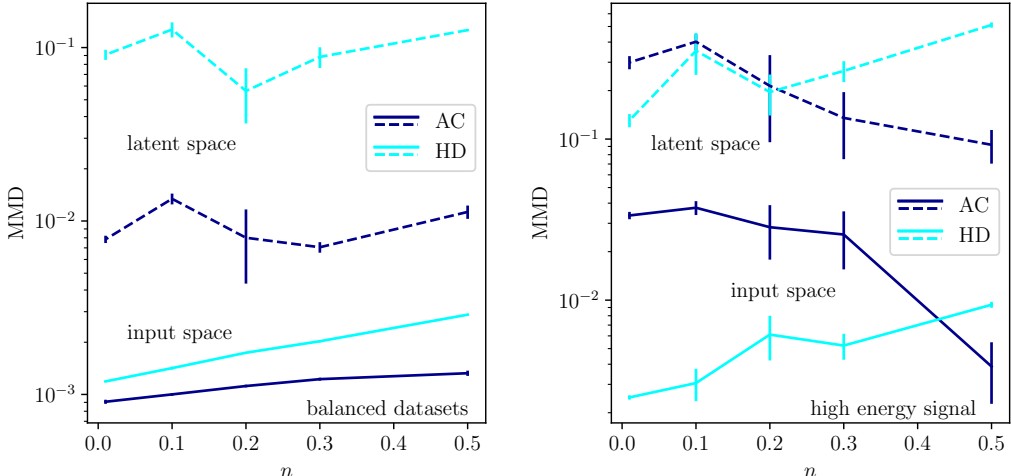

Figure 8: MMD in phase (solid) and latent (dashed) space between two random sample of QCD and signal jets (left), and between the QCD sample and the most poorly reconstructed signal jets (right). Unlike for the other figures, the remapping $n$ increases from left to right.

Moving on to the performance of the NAE on dark jets, we show the ROC curves with different reweightings in Fig. 9. First, we see that the AUCs for the Aachen and Heidelberg datasets are roughly similar. For the sparse Aachen jets we already know that smaller values of $n$ benefit the tagging performance, but we also see that for $n < 0.3$ the AUC reaches values above 0.72, and for $n = 0.2 \ldots 0.01$ the performance essentially plateaus at a high level. In contrast, for the Heidelberg signal we expect a better tagging performance around $\epsilon_S \sim 0.2$ for larger $n$-values.

From Fig.7 we know that the different reweightings mostly change the ordering of the two signal tails at high energies and leave the bulks of the distributions unchanged. The corresponding ROC curves in Fig. 9 confirm that the remaining $n$-dependence is connected to a behavioral change in the model in the region $n \sim 0.2$. While the choice $n = 0.2$ is not optimal for each of the signals, it can be used as a working compromise between sparse dark jets and dark jets related to a mass drop.

## 5 Outlook

Autoencoders are ML-analysis tools which effectively represent the idea behind LHC searches. Unsupervised training can conceptually enrich many aspects of LHC physics, from trigger to analysis techniques. Standard autoencoders identify out-of-distribution jets or events based on the compressibility of their features, a method which is typically not applicable to LHC physics. They are also closely tied to the compressibility of jets or events, a bias we need to avoid because new physics can be more or less complex than QCD. In practice this means that autoencoders should identify backgrounds and anomalies symmetrically. An alternative strategy to define anomalies is based on regions with low phase space densities. The main goal of this work is to define an autoencoder which can reliably identify anomalous jets with an improved training procedure. However, density-based autoencoders by definition have a dependence on data preprocessing [15]. The additional components of the NAE architecture do not increase the size of the network or the inference time, they only increase the time taken to train the model.

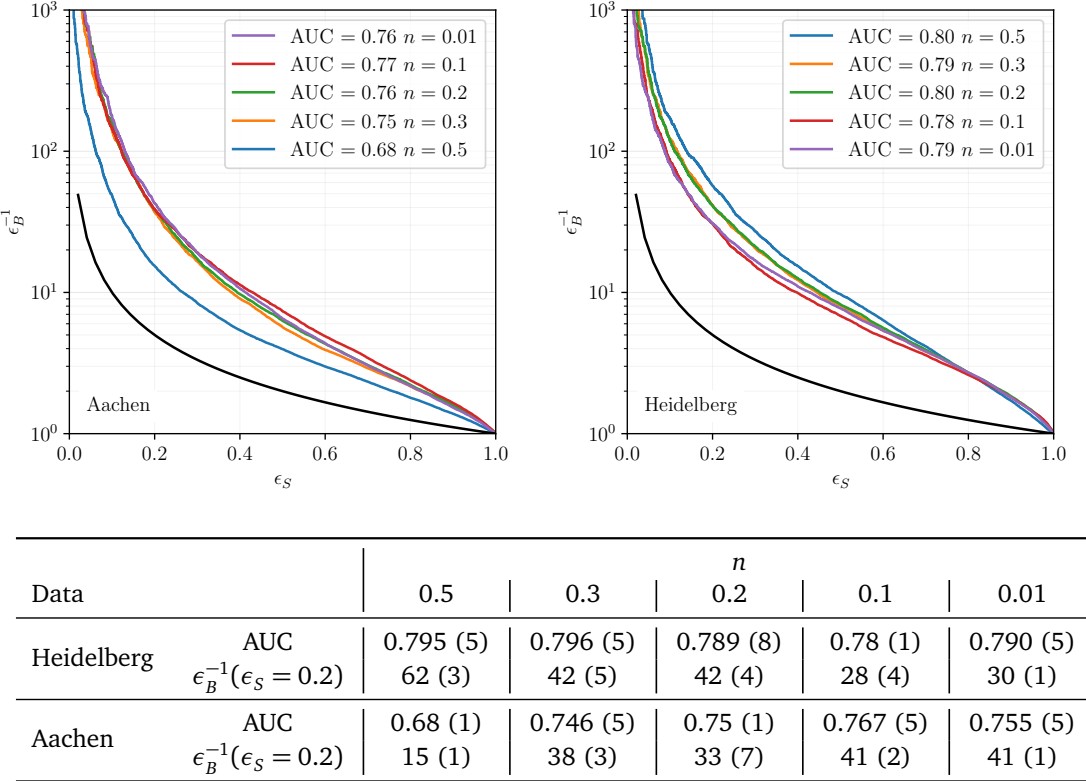

| Data | | n | | | | |
|---|---|---|---|---|---|---|
| | | 0.5 | 0.3 | 0.2 | 0.1 | 0.01 |
| Heidelberg | AUC | 0.795 (5) | 0.796 (5) | 0.789 (8) | 0.78 (1) | 0.790 (5) |
| | $\epsilon_B^{-1}(\epsilon_S = 0.2)$ | 62 (3) | 42 (5) | 42 (4) | 28 (4) | 30 (1) |
| Aachen | AUC | 0.68 (1) | 0.746 (5) | 0.75 (1) | 0.767 (5) | 0.755 (5) |
| | $\epsilon_B^{-1}(\epsilon_S = 0.2)$ | 15 (1) | 38 (3) | 33 (7) | 41 (2) | 41 (1) |

Figure 9: ROC curve for dark jets tagging with different reweightings $n$, shown for the Aachen signal (left) and the Heidelberg signal (right). A random classifier corresponds to the solid black line. The table is based on the same information and shows the mean and the standard deviation of five different runs.

The NAE combines a standard autoencoder architecture with an energy-based normalization in the loss. This means it constructs an energy or MSE landscape such that any anomaly with features not present in the training data will be pushed to even larger energies. Because of the normalization, the NAE can also balance different kinds of features, which means that also the absence of a background feature in signal jets will be visible in the energy landscape. Working on a compact latent space, the NAE can be understood as an extrapolation of the energy distribution beyond the regions defined by the background sample.

Technically, the NAE is just a simple, small autoencoder network. All we adjust is the training after an AE pre-training step. Applied to top vs QCD jets we first show that for this extreme case of different compressibilities the NAE still tags complex top jets in a simple QCD background as well as simple QCD jets in a complex QCD background. In addition, the NAE beats a standard AE as well as the advanced DVAE in top tagging performance.

For the more challenging Aachen and Heidelberg dark jets the NAE works for a reasonable single choice of preprocessing. The performance gain from using different preprocessings on the two datasets is explainable in terms of the changes induced on the features of the two signals. We did not pursue this option further, but we see that it should be possible to construct a latent-space anomaly score for the NAE. While our studies indicate that the NAE is the best-performing autoencoder to date, our setup is optimized for tests and visualization. This means we use a 3-dimensional latent space with a 2-dimensional sphere to track and illustrate the NAE training progress. The next step will be to benchmark our architecture on more realistic datasets, optimize it for performace, and study the possibility of implementing the network on hardware for online triggering.

# Acknowledgments

First, we would like to thank Ullrich Köthe for many inspiring discussions on neural network architectures. We are also grateful to the Mainz Institute for Theoretical Physics, where this paper was finalized. LF would like to thank Sangwoong Yoon for his constant and reliable support.

**Funding information**   This research is supported by the Deutsche Forschungsgemeinschaft (DFG, German Research Foundation) under grant 396021762 – TRR 257: *Particle Physics Phenomenology after the Higgs Discovery* and through Germany's Excellence Strategy EXC 2181/1 - 390900948 (the Heidelberg STRUCTURES Excellence Cluster).

# A   Appendix

## Network architecture

In this section we provide the details of the network architecture and the parameters setting. Tab. 2 summarizes the layers used in both encoder and decoder and their parameters. The layers are defined as:

- Conv2d(`in_ch`, `out_ch`, `filter_size`, `stride`, `padding`, `bias`): 2d convolutional layer;

- Deconv2d(`in_ch`, `out_ch`, `filter_size`, `stride`, `padding`, `bias`): 2d transposed convolutional layer;

- MaxPool2d(`filter_size`, `stride`): 2d max-pooling layer;

- PReLU: rectified linear unit with learnable slope;

- Dense(`in_nodes`, `out_nodes`, `bias`): fully connected layer;

- Upsample(`scale_factor`, `mode`): upsampling layer with bilinear algorithm;

- Flatten: flattens the input into a single dimension;

- Sigmoid: sigmoid activation function.

Table 2: Network architecture. The latent space is $D_{\mathbf{z}} = 3$.

| | |
|---|---|
| Encoder | Conv2d(1, 8, 3, 1, 1, True) - PReLU - <br> Conv2d(8, 8, 3, 1, 1, True) - PReLU - MaxPool2d(2, 2) - <br> Conv2d(8, 8, 3, 1, 1, True) - PReLU - <br> Conv2d(8, 8, 3, 1, 1, True) - PReLU - <br> Conv2d(8, 1, 3, 1, 1, True) - PReLU - Flatten - <br> Dense(400, 100, True) - PReLU - Dense(100, $D_{\mathbf{z}}$, True) |
| Decoder | Dense($D_{\mathbf{z}}$, 100, True) - PReLU - Dense(100, 400, True) - PReLU - <br> Reshape(1, 20, 20) - Deconv2d(1, 8, 3, 1, 1, True) - PReLU - <br> Deconv2d(8, 8, 3, 1, 1, True) - PReLU - <br> Upsampling(2, 'b') - Deconv2d(8, 8, 3, 1, 1, True) - PReLU - <br> Deconv2d(8, 1, 3, 1, 1, True) - Sigmoid |

The architecture of the AE is summarized in Tab. 2. Each layer is regularized using Spectral Normalization. The output of the last encoder layer is mapped to the surface of a hyper-sphere $\mathbb{S}^{D_z-1}$. During training, each step of the preliminary LMC is projected on the surface. The initial latent distribution is uniform but a buffer of size 10000 is used to store the final points of each chain. Then, the initial points are sampled from the uniform distribution with probability 0.05 or from the buffer with probability 0.95.

Additional regularization terms are used to improve training stability. The L2 norm of the weights for both encoder and decoder is added to the loss function with a coefficient $10^{-8}$. We also prevent the negative energy divergence by adding the average squared energy of the training batch to the final loss function.

The bottleneck of the training procedure is the sampling algorithm. The parameters of the LMCs have been tuned to give fine samples after a small amount of steps to train a model in less than 15 hours. We have found that the structure of the initial manifold after pre-training plays an important role for the following NAE updates. A large batch size gave the best results while a smaller one during NAE showed more stable results.

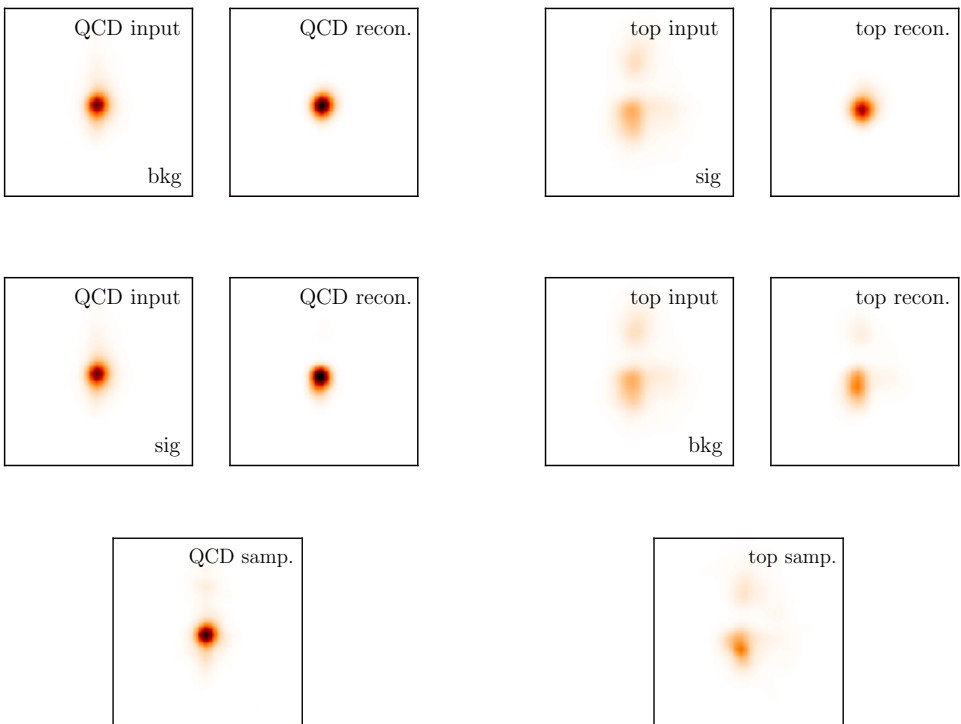

Figure 10: Average of various jet images. The first two rows account for the direct and the inverse tagging problem, and they are showing the average of 10k input and reconstructed images. The last row show an average of 1000 images sampled via LMC when training on the two different backgrounds.

## Jets reconstruction and sampling

The NAE architecture allows us to check explicitly the reconstruction of different jets. Here, we show the average of two subsamples of QCD and top jets with their reconstruction. By comparing the input images and the reconstruction we get a better understanding of the main features learned by the network. These subsamples are shown in Fig. 10. The reconstruction of tops events as signal shows how the network is only able to reconstruct what's in the training

distribution and therefore ignores additional prongs. In the inverse direction, the network detects all three prongs while also wrongly reconstructing QCD signal images. In the latter case the main contribution to the energy is coming from the intensity of the pixels rather than the location of the main prong.

Furthermore, we can explicitly look at sampled images and compare the average distribution to the expected one. The last two images of Fig. 10 show the average sampled distribution for QCD and top tagging. The averaging is performed on 1000 images sampled after training. In both cases the model distribution has converged and resembles the training background. A subset of the LMC samples is shown in Fig. 11.

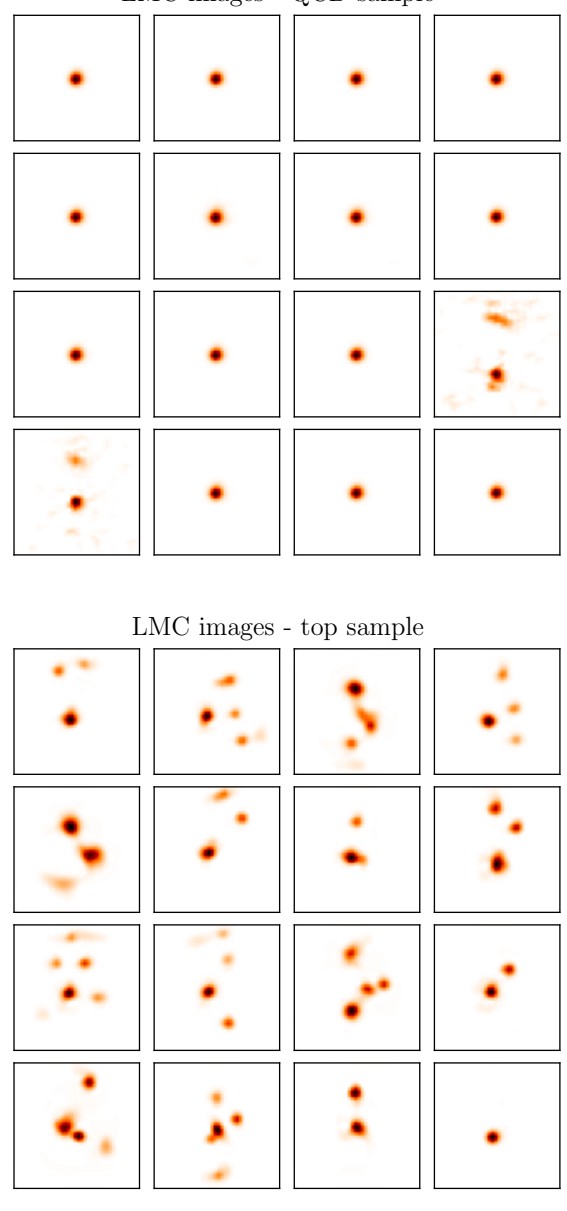

Figure 11: A subset of LMC samples for top (upper) and QCD (lower) tagging.

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
