# Peer review of "A Normalized Autoencoder for LHC Triggers"

_SciPost Physics, doi:SciPost Phys. Core 6, 074 (2023)_

## Round 1 · Referee Report · Anonymous (Referee 1) · 2022-12-31

Report

This paper proposes a new application of unsupervised machine learning to anomaly detection at the LHC. In particular, the Normalized Autoencoder is introduced and studied in a few cases. The topic is important/timely, the work presented in the paper is serious, and the manuscript is well written. I would be happy to recommend publication in SciPost Physics following some comments below (one major and the rest minor).

Major

  • It seems like the main motivation for this work is a method that is symmetric in the following sense: if I have datasets A and B, then p_A(x) is small if and only if p_B(x) is small. I'm not sure this is a necesary motivation and I do not fully understand it. The authors claim that it is a problem that other methods find that QCD jets do not have a low p_top while top jets have a low p_QCD. I don't see the issue - it could be that top jets look much less like a typical QCD jet than QCD jets look like typical top jets. In fact, since the NAE is symmetric, I am suspicious that it has not found the true data probability density.

Minor

Abstract:

  • "Autoencoders are the ideal analysis tool for the LHC" -> "Autoencoders are an effective analysis tool for the LHC"? (see also a similar line in the Outlook)
  • "its main goal of finding physics beyond the Standard Model" -> "one of its main"?

Introduction:

  • "that no assumption should" -> "that few assumptions should" ? Even in your case, you have a preselection and use certain features and these choices build in some assumptions.
  • "The problem with these studies is that it is not clear what the anomalous property of jets or events actually means...." -> I did not understand this paragraph. If a compression algorithm is doing its job, shouldn't it tag as anomalous events with low density? (e.g. should assigm more capacity to common events and less capcity = poor reconstruction to rare events?)
  • "which is smaller enough to run on an LHC trigger" -> I don't think you actually address this? (see also a similar line in the outlook)
  • "Although the jets we study in the paper would pass trigger selection cuts already, the results still demonstrate the utility of this technique on a trigger." -> seems to contradict itself?

Network and dataset:

  • "By using the reconstruction error as the energy, the model will learn to poorly reconstruct inputs not in the training distribution" -> this is perhaps the most important paragraph in the whole paper and I think it could be improved for clarity. Do I understand that the point you are making is that the NAE training sees examples from the model as well as from the data, so it has some "experience" with out of distribution examples (e.g. when the model starts far from the data)? If that is correct, why not try to do this more directly instead of relying on the model being bad? I'm sorry if I have not understood correctly (and in that case, please help me by rewording!)
  • Preprocessing: does this remove important/useful physics information? (I know you study the performance for n later, but this is unsupervised, so it is pospsible that performance gets better/worse "by accident"?) See e.g. https://arxiv.org/abs/1511.05190.

QCD vs top jets:

  • Fig. 4: black line is unlabeled (presumably, it is the random line).

Outlook:

  • "However, density-based autoencoders have not been shown to work properly and have a massive depen- dence on data preprocessing" -> citation?

---

## Round 1 · Referee Report · Anonymous (Referee 2) · 2023-2-13

Strengths

1-well-written pedagogic introduction to autoencoders
2-first (to the best of my knowledge) practical application to jet physics and QCD

Weaknesses

1-examples are too simple

Report

Given the long time this paper had been stalled I suggest publication.

Requested changes

None

---

## Round 2 · Referee Report · Anonymous (Referee 1) · 2023-3-30

Report

Thank you for taking into account my feedback! Overall, I still have some concerns about some of the claims made in the manuscript and I would encourage the authors to rephrase/tone down some of them (see below). However, I will not insist on further changes and I defer to the editor/authors. This paper is an interesting study and I think SciPost Physics is a good venue for presenting the work.

I don't completely agree about you response to my question about symmetry, but I won't insist on further changes/studies. I also don't understand why your autoencoder is not also sensitive to pre-processing. If your method is learning something about the underlying density of the data (as all compression models do) then it should also be sensitive to how you pre-process the data. Some of the claims along these lines seem too strong.

I still find the strong emphasis on triggers (even in the title!) to be unjustified, as there is really no evidence that your approach is trigger friendly or is otherwise specifically designed for running online. It is true that other people have shown that AEs can be implemented in hardware/firmware, but that I don't see how that justifies such a big claim for your paper. Please consider further modifications along these lines.

Please change the first line of the conclusions ("Autoencoders are ML-analysis tools which ideally represent the idea behind LHC searches.") along the changes you made in the abstract/introduction.

---

## Round 2 · Author Response

We appreciate the feedback from the referees. We reviewed the manuscript based on the reports. We discuss the major and minor issues below.

We would like to point out that the detailed report by the first referee suggests the publication in Scipost Physics.
We think that the introduction of Normalized AutoEncoders is an important step towards necessary anomaly detection searches with robust autoencoders in Particle Physics. Therefore, we believe the paper should be published in Scipost Physics.

---

## Round 2 · List of Changes

Changes based on report 1:

Major issue: We agree with the referee that, given two datasets A and B, an over-density in the low probability region p_A doesn't imply an over-density in the low p_B. However, the underlying physics suggests this is true when comparing QCD and top jets. A top jet will likely decay (> 70% of the time) to a three prong jet, therefore it is expected that a large region of phase space will be populated exclusively by QCD jets which must look differently. The differences between the average images of QCD and top jets and in high-level observables (e.g. tau_3/tau_2) corroborate this point. Given these differences, it is indeed surprising that an AE is not able to tag QCD jets, and we argue that this is connected to the ability of the network to interpolate simple features. Additionally, it has been shown that the complexity bias is an issue for AE since reconstruction heavily correlates with the number of active pixels (e.g. see arXiv:2104.09051).

In this work, we show that an NAE is symmetric in the sense that we can reliably find overdensities in low-probability regions without modifying the network architecture and fine-tuning the parameters. We see the difference between performing density estimation on the direct or inverse task, as shown in Figs. 3-4, so there's no indication that we are not approximating the true data distribution. We rephrased the first paragraph of the "QCD vs top jets" section to make this clear.

Minor changes: - abstract: "Autoencoders are the ideal analysis tool for the LHC" -> "Autoencoders are an effective analysis tool for the LHC" "its main goal of finding physics beyond the Standard Model" -> "one of its main"

  • introduction: "that no assumption" -> "that as few assumptions as possible" "The problem with these studies...": The end of the paragraph addresses this point. QCD jets are easier to compress since they have a simple structure. In other words, the effective dimensionality of a QCD jet is on average smaller than a top jet. Therefore, the network is still able to interpolate these features. We modified the corresponding paragraph to explain our claims on applications of NAEs for LHC triggers.

  • network and dataset: "by using the reconstruction error ..." -> "The NAE training includes two steps... The pre-training phase builds an approximate..." A new paragraph describes the training procedure and the role of the modified loss function.

-Preprocessing: We do not expect to lose physical information. The picked preprocessing does not modify the structure of the jet (e.g. prongness). Additionally, n-reweighting has been selected to enhance features in jets with low complexity. Therefore, the improvement found in our dark jets example is not an accident.

  • QCD vs top and QCD vs dark jets: added black line label in captions

  • Outlook: "However, density-based autoencoders have not been shown to work properly and have a massive dependence on data preprocessing" rephrased and added citations.

---

## Round 3 · Referee Report · Anonymous · 2023-9-27

Report

Thank you to the authors for their patience in this prolonged review process. At this point, I am fine with publishing the current version in SciPost.

The only change I would strongly suggest to the authors is to consider changing the title and any mention of "trigger". From the last response, it sounds like the authors really don't mean "trigger" as it is usually meant at the LHC. You say "not necessarily hardware/online triggers
but tools to extract 'interesting' events. ". If it is not online, then it is not a trigger! *Tagging* anomalous events is probably a better way of describing your work. Otherwise, I fear that the title is rather misleading.

---

## Round 3 · Author Response

We thank the referee for the feedback. We carefully reviewed the manuscript and rephrased statements on the effect of preprocessing and trigger applications.
However, we are not stating that our network is insensitive to changes in the probability distribution (preprocessing).
In Section 4, we discuss the changes induced by the reweighting in the data space and the response of the NAE samples and reconstructions.
We provide clarification on statements about preprocessing below. Please let us know if your concerns arise from more specific claims.

---

## Round 3 · List of Changes

Preprocessing:
* * *
- Although the jets we study in the paper would pass trigger selection cuts already,
the results still demonstrate how our approach limits the assumptions
made on BSM signals to data preprocessing rather than latent space structure,
in favor of a more model-agnostic network architecture

- For phase space
regions with such modes the NAE training adjusts the energy as the
underlying structure of the latent space, such that the autoencoder
gets a robust OOD detector

- This way, the NAE training increases the energy for
Aachen jets, because the normalization forces the main prong to a
lower value. These effects are inevitable when training likelihood-based networks,
a different preprocessing will change the density and, therefore, the anomaly scores.

- Once again focusing on the $p_T$-reweightings we show the energy
distributions for the QCD training data and the two signals in Fig.~\ref{fig:djmse}.
We see that unlike in our earlier study~\cite{Buss:2022lxw} the effect of the preprocessing
on the whole distribution is limited. A shift in performance at low signal
efficiency can be seen by varying $n$ with the ordering between
the two dataset being switched around $n = 0.2~...~0.3$.
The energy distribution of the Heidelberg dataset has a shifted
main peak at $n=0.5$ which is washed out by smaller reweighting factors,
while the QCD distribution undergoes a slight shift and develops a
longer high-energy tail. For the Aachen dataset, lowering $n$ moves the mean away
from the QCD background and at the same time increases the width of the distribution.
These patterns will affect the ROC curves at low signal efficiency and large background
suppression.

- Removed: However, already looking at the AUC as a performance
measure this changes, because the performance ordering as a function
of $n$ changes towards larger signal efficiencies.

- For the more challenging Aachen and Heidelberg dark jets, the NAE works
for a reasonable single choice of preprocessing. The performance gain
from using different reweighting factors on the two datasets is explainable
in terms of the changes induced on the features of the two signals.

- Autoencoders are ML-analysis tools which effectively represent the idea
behind LHC searches.
* * *
Triggers:
Highlight using autoencoders as of the idea of triggers: not necessarily hardware/online triggers
but tools to extract "interesting" events. Left hardware applications for future work.

- One of the goals of this work is to develop an autoencoder which is a robust anomalous jets tagger.
We explore the concept of using autoencoders as of triggers, i.e. tools that can extract
interesting events from a given background with as little bias as possible.

- The next step will be to benchmark our architecture on more
realistic datasets, optimize it for performace, and study the
possibility of implementing the network on hardware for
online triggering.

---

## Editorial Decision

published